# Interest without uptake: A mixed-methods analysis of methadone utilization in Kyrgyz prisons

Amanda R. Liberman[1]*, Daniel J. Bromberg[2,3], Taylor Litz[1¤], Ainura Kurmanalieva[4], Samy Galvez[1,3], Julia Rozanova[1], Lyu Azbel[1], Jaimie P. Meyer[1,3], Frederick L. Altice[1,3]

**1** Yale University School of Medicine, New Haven, CT, United States of America, **2** Yale University School of Public Health, New Haven, CT, United States of America, **3** Yale Center for Interdisciplinary Research on AIDS, New Haven, CT, United States of America, **4** AIDS Foundation East-West (AFEW) in the Kyrgyz Republic, Bishkek, Kyrgyz Republic

¤ Current address: Douglas County Health Department, Omaha, NE, United States of America
* amanda.liberman@yale.edu

**Data Availability Statement:** All relevant quantitative data are within the paper and its Supporting Information files. Data have been deidentified. Individual demographic information

## Abstract

HIV incidence continues to increase in Eastern Europe and Central Asia (EECA), in large part due to non-sterile injection drug use, especially within prisons. Therefore, medication-assisted therapy with opioid agonists is an evidence-based HIV-prevention strategy. The Kyrgyz Republic offers methadone within its prison system, but uptake remains low. Screening, Brief Intervention, and Referral to Treatment (SBIRT) is a framework for identifying people who would potentially benefit from methadone, intervening to identify OUD as a problem and methadone as a potential solution, and providing referral to methadone treatment. Using an SBIRT framework, we screened for OUD in Kyrgyz prisons among people who were within six months of returning to the community ($n$ = 1118). We enrolled 125 people with OUD in this study, 102 of whom were not already engaged in methadone treatment. We conducted a pre-release survey followed by a brief intervention (BI) to address barriers to methadone engagement. Follow-up surveys immediately after the intervention and at 1 month, 3 months, and 6 months after prison release assessed methadone attitudes and uptake. In-depth qualitative interviews with 12 participants explored factors influencing methadone utilization during and after incarceration. Nearly all participants indicated favorable attitudes toward methadone both before and after intervention in surveys; however, interest in initiating methadone treatment remained very low both before and after the BI. Qualitative findings identified five factors that negatively influence methadone uptake, despite expressed positive attitudes toward methadone: (1) interpersonal relationships, (2) interactions with the criminal justice system, (3) logistical concerns, (4) criminal subculture, and (5) health-related concerns.

has not been shared to prevent potential participant identification; it will be available upon request to the Yale Center for Interdisciplinary Research on AIDS (contact via data manager Delaney Rhoades, delaney.rhoades@yale.edu) for researchers who meet the criteria for access to confidential data. Full interview transcript data cannot be shared publicly because of potential identifiability of our participants. Qualitative data are available from the Yale Center for Interdisciplinary Research on AIDS (contact via data manager Delaney Rhoades, delaney.rhoades@yale.edu) for researchers who meet the criteria for access to confidential data.

**Funding:** Funding for this study was provided by the National Institute for Drug Abuse (NIDA, drugabuse.gov; K01 DA047194 received by JR, R01 DA029910 received by FLA, & R21 DA042702 received by JPM). DJB's time devoted to this paper was provided by the National Institute for Mental Health (NIMH, nimh.nih.gov; T32 MH020031) and the Fogarty International Center (fic.nih.gov; D43 TW010540). ARL's time devoted to this paper was provided by a G.E.R.M. award from the Infectious Diseases Society of America (IDSA) Foundation (idsafoundation.org/g-e-r-m). The funders had no role in study design, data collection and analysis, decision to publish, or preparation of the manuscript.

**Competing interests:** The authors have declared that no competing interests exist.

## Introduction

Eastern Europe and Central Asia (EECA) is one of the few regions globally where HIV incidence continues to increase [1]. In the Kyrgyz Republic, this increase is largely due to unsafe injection practices, particularly within prisons [1, 2]. Methadone, the most effective treatment for opioid use disorder (OUD) [3, 4], is also highly effective at preventing HIV [5, 6]. While methadone has been available within most Kyrgyz prisons since 2008, uptake among incarcerated people remains low [7, 8]. Previous research has suggested multiple factors for this low uptake. For instance, much has been written about the criminal subculture that governs Kyrgyz men's prisons, influencing nearly every within-prison behavior including whether an individual has access to heroin [9–11]. The frequent use of Dimedrol (diphenhydramine), a soporific, in conjunction with methadone in Kyrgyz prisons means that the effects of Dimedrol are often conflated with those of methadone [12, 13]. Finally, current Kyrgyz Ministry of Health guidelines specify low dosages of methadone (30 mg initial dose, increasing by 5-10mg every 7 days), potentially increasing risk of dropout [14].

Release from prison carries a very high risk of death due to overdose [15, 16], particularly among people living with HIV [12, 15, 17]. Given the generally low levels of methadone uptake in Kyrgyz prisons as well as the increased risks surrounding release, we deployed a screening, brief intervention, and referral to treatment (SBIRT) strategy to increase methadone program participation among incarcerated people with OUD who were scheduled to be released from prison within six months.

SBIRT is an evidence-based strategy to identify people with substance use disorders and engage them in care [18]. It has been deployed in multiple community settings in the US with modest effectiveness [19–24], although to our knowledge, only one other study (from this lab) has examined its effectiveness in another EECA country [25], and it has not previously been implemented in the Kyrgyz Republic. In this study, we recruited 125 soon-to-be released incarcerated people with OUD in the Kyrgyz Republic who participated in the SBIRT intervention. Additionally, we conducted in-depth interviews with 12 participants before and after their release from prison. The resulting analysis allowed us to investigate both interest and uptake in methadone utilization among soon-to-be-released people with OUD in Kyrgyz prisons. Furthermore, it allowed us to explore some of the reasons behind the lack of methadone uptake, despite professed positive attitudes toward this treatment.

## Materials and methods

This study began in October 2016 and remains ongoing in nine prisons in the Kyrgyz Republic. Methods for recruitment have been previously described [13]; ClinicalTrials.gov Identifier for Project MATLINK is NCT04947475. The study used a screening, brief intervention, and referral to treatment (SBIRT) strategy. Briefly, research personnel screened all incarcerated persons between 8 and 180 days from their release date using a single-item screener for opioid use disorder (OUD) followed by the Rapid Opioid Dependence Scale. In total, we recruited 125 people into our sample. Given the utilization of the SBIRT method (i.e., we cannot recruit more people than are eligible participants in the country), a formal sample size/power calculation was not performed.

If OUD was confirmed, potential participants completed informed consent procedures in which research assistants made clear that this study was not affiliated with the prison administration, that surveys and interviews would remain anonymous, that participants could withdraw from the study at any point, and that neither participation nor withdrawal were linked with any rewards or punishments. Ethical approval for the study was provided by the US Department of Health and Human Services, Office for Human Research Protections (OHRP)

and by the institutional review boards (IRBs) at Yale University and at the Global Research Institute Foundation in the Kyrgyz Republic. Yale's IRB included an incarcerated person as a representative.

After enrollment, participants were assessed for initial interest in methadone on a scale from 0 to 10, with 0 indicating no interest in methadone and 10 indicating a plan to begin methadone treatment. Next, participants completed surveys assessing demographic characteristics, OUD severity (using the Addiction Severity Index-Lite [26]), depression (using the CES-D scale [27]), and overall physical and mental health (using the SF-12 [28]). A complete list of survey questions is available (S1 File; results available in S3 File). Participants then underwent testing for HIV, HBV, HCV, and syphilis.

Next, they participated in a brief intervention (BI) guided by motivational interviewing principles in which a trained research assistant explained benefits and dispelled myths relating to methadone treatment both during and after incarceration. The BI informed participants of the risks of substance misuse by illustrating potential adverse health consequences. Additionally, the BI aimed to motivate participants to seek treatment for their substance use disorder. After the BI, participants' interest in methadone was re-assessed, and if interested, they were referred to a treating physician in the prison to initiate methadone. All study participants, irrespective of methadone enrollment, underwent a second BI one week before release to encourage participants either to initiate methadone treatment or to link to care in the community. Each BI lasted approximately 20 minutes, and afterwards, participants were provided time for questions. The BIs were audio recorded; audio files are available upon request (in the original languages of the BI—Russian & Kyrgyz).

After release, study participants underwent repeat consent procedures and were followed up at 1, 3, and 6 months to assess for methadone interest or uptake or continued opioid use; for 1 individual who did not complete any surveys within 6 months, a 12-month survey was administered. If reincarcerated post-release, study participants were listed as not available for follow-up. Methadone uptake was verified using a state-run methadone registry. To further understand perceptions of methadone in and outside of prison settings, in-depth qualitative interviews were carried out from 12 study participants both pre- and post-release. LA and JR conducted the interviews in Russian. Interviews lasted, on average, 45 minutes and were audio-recorded (See S2 File for interview guide).

The initial coding of these interviews is discussed in a previous paper [13]. For the present paper, the authors used thematic analysis based on risk environment theory to sub-code the data, looking for factors relevant to methadone uptake and utilization before and after release from prison. In this text, information is provided about participants' levels in the within-prison social hierarchy. An individual can be classified into one of three general categories, in descending order of status: 1) *poryadochnyi* ("decent one"); 2) *neput'* ("one who lost the way"); 3) *obizhennyi* ("one who has offended"), and one can be promoted or demoted based on various behaviors or interactions with people in other levels of the hierarchy, as described in previous studies [29, 30].

## Statistical analysis

Statistical analysis was completed in Microsoft Excel and R. Given that Shapiro-Wilks tests indicated non-normality for all survey questions ($p<0.001$ for all), paired, one-sided non-parametric (Wilcoxon signed rank) tests were used to compare survey scores at baseline vs. at follow-up (alternative hypothesis: "Survey outcome increased post-intervention"). Participants who had not completed both a baseline and a follow-up survey were excluded from analysis. Primary study outcomes included initiation of methadone, retention in methadone treatment, relapse to heroin, and recidivism.

## Results

### Findings from the screening, brief intervention and referral to treatment strategy

Between 2016 and 2021, 1,118 soon-to-be-released incarcerated people underwent screening, and 125 (11.2%) screened positive for OUD and enrolled in the study (Fig 1). Study participant characteristics are available in Tables 1 and 2. While information is not available for all of these characteristics for the prison population of the Kyrgyz Republic, the percentage of female

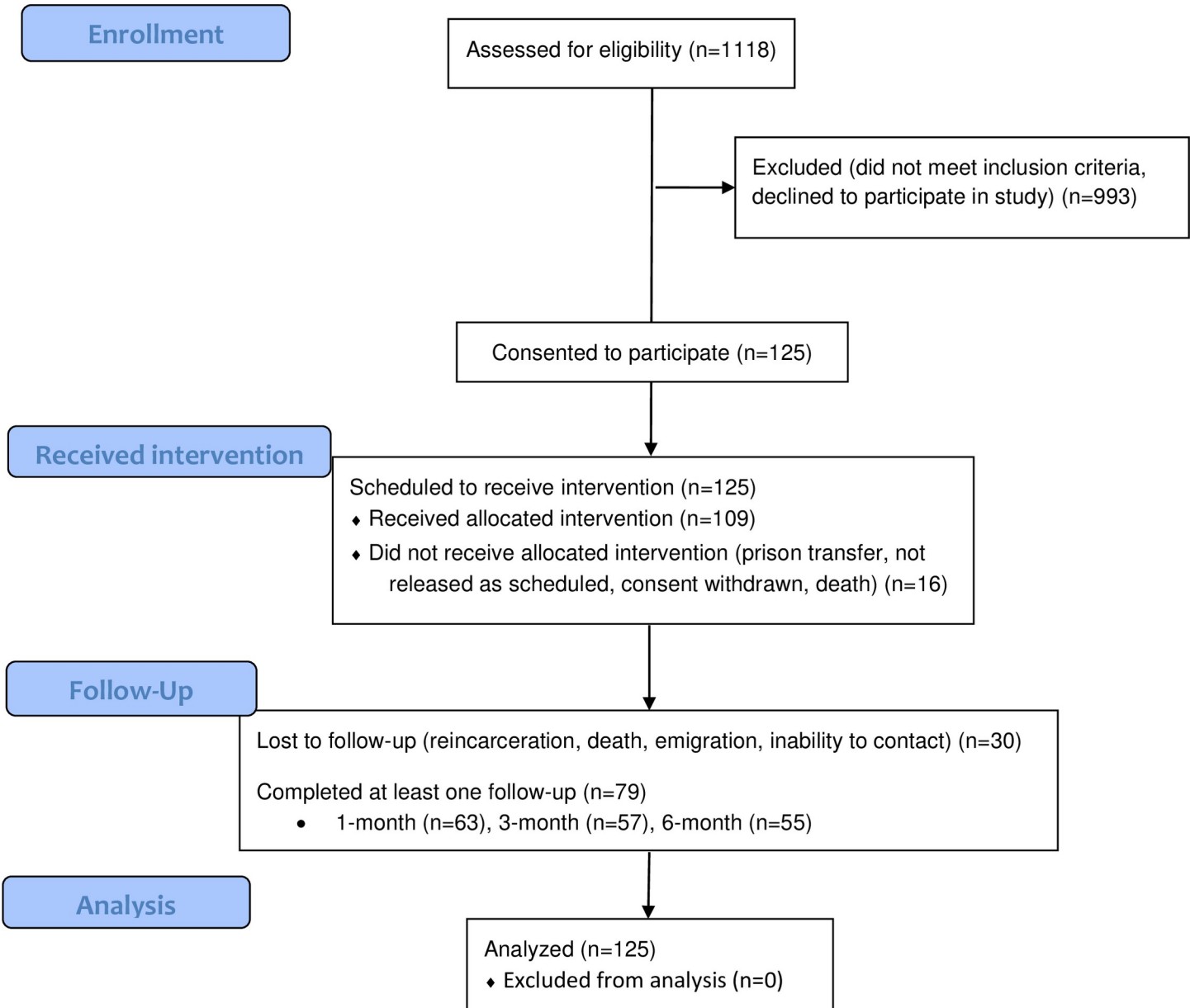

**Fig 1. Modified CONSORT [32] flow diagram.** 1,118 incarcerated people scheduled to be released within 6 months were screened for opioid use disorder, and 125 screened positive and consented to participate in the study. Of those, 109 completed the full pre-release visit questionnaire, 63 followed up 1 month post release, 57 followed up at 3 months, and 55 followed up at 6 months. Note that some participants did not complete one follow-up visit but returned later in the study; for example, not all of the 55 participants who completed 6-month follow-up had also completed the 3-month follow-up.

**Table 1. Survey participant characteristics.**

| Variable | N | % | Mean | sd | Range |
|---|---|---|---|---|---|
| *Sex* | | | | | |
| Male | 104 | 92.9 | | | |
| Female | 8 | 7.1 | | | |
| *Age* | | | 39.8 | 8.4 | (24, 66) |
| *Ethnicity* | | | | | |
| Kyrgyz | 28 | 24.8 | | | |
| Russian | 57 | 50.4 | | | |
| Uzbek | 5 | 4.4 | | | |
| Other | 23 | 20.4 | | | |
| *Marital Status* | | | | | |
| Partnered | 46 | 40.7 | | | |
| Not partnered | 67 | 59.3 | | | |
| *Education* | | | | | |
| Secondary or less | 108 | 95.6 | | | |
| Beyond secondary | 5 | 4.4 | | | |
| *Housing* | | | | | |
| Self-provided residence (rent/own) | 19 | 16.8 | | | |
| Friend or Relative's home | 71 | 62.8 | | | |
| Other | 23 | 20.4 | | | |
| *Employment* | | | | | |
| Full/Part-time employment | 37 | 58.7 | | | |
| Unemployed | 26 | 41.3 | | | |
| *HIV* | | | | | |
| Preliminary Positive | 25 | 22.1 | | | |
| Preliminary Negative | 88 | 77.9 | | | |
| *Hepatitis B* | | | | | |
| Preliminary Positive | 6 | 5.3 | | | |
| Preliminary Negative | 108 | 94.7 | | | |
| *Hepatitis C* | | | | | |
| Preliminary Positive | 109 | 95.6 | | | |
| Preliminary Negative | 5 | 4.4 | | | |
| *Syphilis* | | | | | |
| Preliminary Positive | 5 | 4.4 | | | |
| Preliminary Negative | 109 | 95.6 | | | |
| *Depression (CESD-10)* | | | | | |
| Yes | 68 | 60.7 | | | |
| No | 44 | 39.3 | | | |

This table describes survey participant characteristics (*n* = 125). However, only 117 participants completed at least part of the questionnaire or infectious disease testing, and many participants left some survey answers blank. For ethnicity, "other" includes ethnicities that fewer than 5 people indicated (Azerbaijani, Belarusian, Dungan, German, Kazakh, Kurdish, Moldovan, Tatar, Turkish, Uighur, or Ukrainian ethnicity).

prisoners in our study and high rate of infectious diseases like hepatitis C correlate with findings reported by other sources [8, 31]. Of these 125 study participants, 109 completed the pre-release visit questionnaires and attended the brief intervention (BI), a workshop that used motivational interviewing techniques to address barriers to methadone engagement. Of these 109, 63 participated in a one-month follow-up after release, 57 participated in the three-month

**Table 2. Qualitative interview participant characteristics.**

| Gender | Age Range | Ever in methadone program? (bold if taking methadone at follow-up) | Status in hierarchy (applies to men only) |
|---|---|---|---|
| Male | 41 to 45 | **yes** | *poryadochnyi* |
| Female | 36 to 40 | **yes** | |
| Female | 41 to 45 | no | |
| Male | 31 to 35 | **yes** | *obizhennyi* |
| Male | 46 to 50 | yes | *neput'* |
| Male | 46 to 50 | no | *poryadochnyi* |
| Female | 26 to 30 | yes | |
| Male | 36 to 40 | no | *poryadochnyi* |
| Male | 31 to 35 | no | *poryadochnyi* |
| Male | 41 to 45 | yes | *poryadochnyi* |
| Male | 31 to 35 | no | *poryadochnyi* |
| Male | 41 to 45 | yes | *poryadochnyi* |
| Male | 56 to 60 | **yes** | *poryadochnyi* |

This table describes participant characteristics of interviewees. Hierarchy status refers to the within-prison social hierarchy. These include, in descending order of status: 1) *poryadochnyi* ("decent one"); 2) *neput'* ("one who lost the way"); 3) *obizhennyi* ("one who has offended"). A person could be promoted or demoted based on various behaviors and interactions with people of different hierarchy statuses [29, 30].

follow-up, and 55 participated at six-month follow-up (Fig 1). These post-release follow-ups did not have all the same participants; some participants did not participate for the one-month follow-up but attended the three-month follow-up, for example. Participants who did not complete the study did not differ substantially in demographic characteristics from the wider study population.

Initially, 23 participants were already accessing methadone treatment, and 44 participants expressed interest in initiating methadone treatment. After the BI, 50 (including the 44 named above) expressed interest in initiating treatment; however, only four participants actually initiated treatment (Fig 2). Interest in methadone was assessed on a 10-point Likert scale, and it did not change significantly after the intervention (Table 3, $p$ = 0.135). Additionally, scores on survey questions assessing attitudes toward methadone did not change significantly after the intervention (Table 4). At study initiation, most scores on survey questions were 4 or higher on a 5-point Likert scale—nearly all study participants agreed, for example, that methadone should be available in the community and in prisons. However, these reported positive attitudes toward methadone did not translate into interest, which remained at a median score of 0 out of 10 (Table 3). Of the 50 participants who did express newfound interest in the methadone program following the intervention, only 4 initiated treatment.

## Qualitative interview findings

We turned to qualitative interview data to determine some of the reasons behind the discrepancy between professed positive attitudes toward methadone and lack of methadone uptake. Generally, perception of methadone from in-depth interviews revolved around one of five themes: (1) interpersonal relationships, (2) interactions with the criminal justice system, (3) logistical concerns, (4) criminal subculture, and (5) health-related concerns.

**Interpersonal relationships.** When asked what could be done to increase methadone uptake in the prison, one study participant explained that incarcerated people with OUD faced

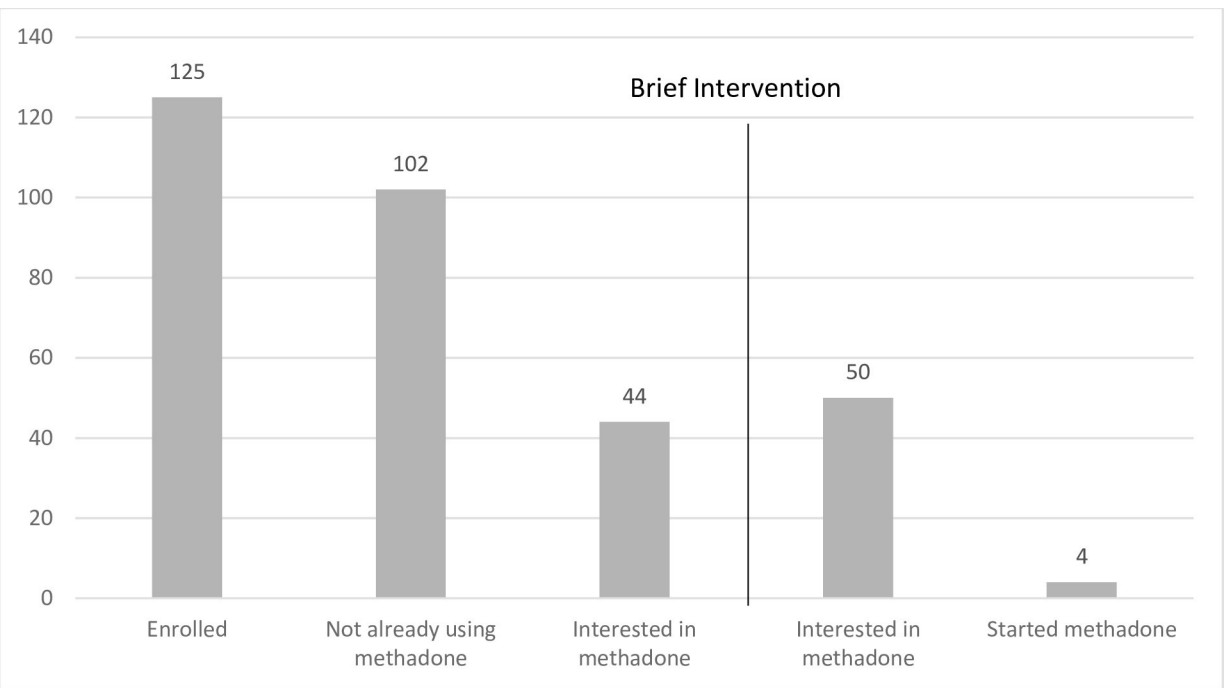

**Fig 2. Methadone interest cascade for incarcerated Kyrgyz study population.** 125 people were initially enrolled in the study, of whom 102 were not already participating in the methadone program. 44 expressed interest before a brief intervention, and 50 expressed interest after ($p$ = 1.00). However, only 4 people joined the methadone program following the intervention.

frequent stigma from medical providers. "*Well, with regard to medical care. . .They sometimes look at us as if we weren't human beings, you understand, and this really affects a person's morale, right?*" (female, 41–45, never on methadone). Another methadone program participant described how his peers in the prison viewed him, explaining that "*. . .it is obvious people have. . . some animosity, some loathing, right, to put it simple.*" (male, 41–45, on methadone). Another participant not on methadone agreed. "*In the prison, normal people won't communicate with those who use methadone.*" (male, 41–45, on methadone).

Fellow incarcerated people tended to describe methadone users as weak or suggestible. One participant explained "*there are people who cannot put up with their pain, they go for [methadone] out of despair, even though it doesn't help them*" (female, 41–45, never on methadone). Another said that if a person starts taking methadone within the prison "*this is called aping. . . 'are you an ape, you saw him doing these moves and now you want [methadone] too,'*" (male, 56–60, on methadone). According to a third, methadone participants are "*just afraid of getting off it. They're afraid of withdrawal.*" (male, 46–50, formerly on methadone).

People on methadone were not considered "sober", which could be a source of stigma. "*You're in a circle of sober people, no one is shooting up, they're against it, right, to shoot up, and you are alone among them. Well, they look at you as if you're an animal*" (male, 31–35, on methadone). As another participant explained, "*my family. . .laid down a condition for me, you get over withdrawal, that's it, come over, we'll help you. . . for them it's all the same thing,*

**Table 3. Interest in methadone before and after brief intervention (n = 109).**

|  | **Before Intervention** | **After Intervention** | ***p*-value** |
|---|---|---|---|
| Interest Score, median (IQR) | 0.0 (0–0) | 0.0 (0–2) | 1.00 |

**Table 4. Mean methadone attitude and knowledge scores at baseline vs. 1-month follow up.**

| Statement | Baseline Mean (SD) | Follow Up Mean (SD) | p-value |
|---|---|---|---|
| 1. Methadone should be available in the community so that all people who suffer from opioid addiction and want methadone can receive it. | 4.37 (0.89) | 4.50 (0.82) | 0.82 |
| 2. Methadone should be introduced into prisons so that all incarcerated people who suffer from opioid addiction and want methadone can receive it. | 4.38 (0.89) | 4.42 (0.92) | 0.57 |
| 3. Methadone reduces opioid dependent individuals' consumption of illicit opiates. | 4.18 (0.95) | 4.27 (0.63) | 0.74 |
| 4. Methadone reduces opioid dependent individuals' risk of acquiring or transmitting HIV. | 4.35 (0.96) | 4.42 (0.86) | 0.63 |
| 5. Methadone improves adherence to HIV medications in HIV-infected opioid dependent individuals. | 3.85 (1.11) | 3.87 (1.09) | 0.64 |
| 6. Methadone increases opioid dependent patients' adherence to tuberculosis medication. | 3.58 (1.19) | 3.35 (1.13) | 0.19 |
| 7. Methadone decreases opioid dependent individuals' risk of dying from overdose. | 4.29 (0.91) | 4.40 (0.64) | 0.74 |
| 8. Methadone reduces addicts' criminal activities. | 4.37 (0.89) | 4.29 (0.88) | 0.29 |

Mean methadone attitude and knowledge scores at baseline vs. 1-month follow up ($n = 63$). Range of responses is 1–5 (1 = strongly disagree, 5 = strongly agree).

*whether methadone, heroin, or cocaine, it's all drugs*" (male, 56–60, on methadone). Often study participants had the same beliefs and quest for "sobriety". When one participant expressed her desire to leave the methadone program, she viewed medical professionals' discouragement as trying to hold her back from her full potential. "*Because if I'm sober, I'm a respected person everywhere. . . everybody will respect me, and this is what alarms them*" (female, 36–40, on methadone). As another participant stated, "*I think only a person who doesn't want to get sober will go on methadone*" (male, 46–50, on methadone).

While young people faced numerous social consequences for methadone program participation, among older incarcerated people, methadone was seen as unavoidable. "*I told the deputy right away, I've been drinking it for so many years, my bones are all soaked with it, with methadone, that's why I'm not even thinking about quitting and I won't ever quit. Well, that was it, the conversation was over*" (male, 56–60, on methadone). However, these older individuals would strongly discourage younger, newly incarcerated people from entering into the methadone program and would urge them to take up sports or athletic pursuits instead.

**Interactions with the criminal justice system.** Participants described police as being generally suspicious of people taking methadone in the community, often assuming they were swindling the system to get "free drugs." This was especially difficult for younger people; police usually left older people alone. Police officers would congregate near methadone programs and arrest people, often framing them for other crimes in the process.

In nearly all interviews, participants described the police as enabling heroin use in the community and framing those who used heroin for other crimes. As one participant explained, "*The cops themselves got me hooked on heroin, so that I would work and split [my profits] with them*" (female, 36–40, on methadone). As another participant explained, "*if my health really deteriorates. . .it's better to [take] methadone, only based on the fact that with heroin I'll go back to prison, they'll make me admit to someone else's crimes, they'll write me down as a contraband dealer again, although I've never been a contraband dealer in my life*" (male, 46–50, never on methadone). Generally, all interviewees were distrustful of the police and prison systems, and this distrust often extended to the methadone program.

**Logistics of taking methadone.** Logistical concerns about taking methadone differed within prison and after release. Within prison, concerns included dilution of methadone and loss of access to heroin. As one interviewee explained, *"Sometimes they add some water. When the water is added. . . Of course, we [notice]. And those who don't feel the difference–they are not real drug users"* (female, 26–30, formerly on methadone). Once someone entered the methadone program, they were no longer provided access to the informal within-prison heroin distribution network or to the administration-run needle/syringe program (NSP). Ordinarily, to access the NSP, individuals registered confidentially and were then able to receive injection equipment from nurses located either in the medical area or the barracks, depending on the specific prison. However, those that joined the methadone program were no longer permitted to participate in the NSP. Loss of access to the NSP could be a major life change and was a dealbreaker for some would-be methadone program participants. Those unwilling to give up heroin completely would shy away from the methadone program; some people would leave the methadone program after realizing that they had lost access to heroin and the NSP.

Upon release, incarcerated people were often not provided referrals or tools to find a methadone program in their region, since methadone is only available in certain regions of the Kyrgyz Republic. Additionally, community methadone programs required being tied to a daily clinic, and some participants reported that their program would not allow them to return if they missed a day of their methadone dose. These logistical concerns made accessing methadone in the community difficult, especially for people whose jobs conflicted with the set methadone distribution times.

**Criminal subculture.** Within Kyrgyz men's prisons, a strict hierarchy system governs daily life among incarcerated people. This hierarchy, run by an incarcerated-person-led government called the *obshchak*, has been described previously [29, 30]. Briefly, when a person arrives into prison, he encounters a tribunal of his peers, which assesses whether he is guilty of the crime for which he has been incarcerated, the severity of his crime, and any mitigating factors (for example, positive character references from community members). He is then assigned a hierarchy status based on this assessment (Table 2).

For someone of high hierarchy status, there was little motivation to join the methadone program, and methadone carried social risks. Meanwhile, for someone low in the hierarchy, who was largely denied access to within-prison heroin and had nothing to lose in terms of social status, the methadone program was much more appealing. One high-status person explained his fears about having to choose between hierarchy status and methadone program participation. When one of his friends joined the program, the friend was quickly approached by *obshchak* enforcers. *"'If you proceed with methadone, we will relocate you and everything you have now, you will lose it. Well, your quality of life would change, get it?'"* (male, 41–45, on methadone). Prison medical staff administered methadone daily at a specific, designated location for all methadone program participants irrespective of hierarchy level. Therefore, for someone of higher hierarchy status, joining the methadone program meant potential physical interaction with people of lower hierarchy status or using the same items, like pens or cups. These interactions could lead to demotion within the hierarchy [29].

The *obshchak* was highly motivated to dissuade people from using methadone, because the *obshchak* was the major distributer of in-prison heroin. This process was facilitated by the *obshchak*'s extensive connections outside of the prison. It acted both as a mutual aid fund, collecting and redistributing goods to incarcerated people, and also as a marketplace for various goods and services. This marketplace was facilitated by corruption of official prison staff, who allowed these goods, including heroin, to enter the prison [11]. Heroin served both as a commodity and as a form of currency which could be used to purchase other items within the prison. Incarcerated people could work for the *obshchak* in exchange for heroin, so methadone

uptake resulted in net economic losses for the *obshchak*. However, some methadone program participants would continue to work for the *obshchak* as a way of maintaining access to heroin for bartering, although they were absolutely forbidden from keeping any of that heroin for themselves.

Despite its role as heroin provider, many incarcerated people described the *obshchak* as an ally in the quest for "sobriety". *"[The* obshchak*] will even help him to quit this methadone. You have strong withdrawal from methadone, and so that he doesn't get strong withdrawal, they give him, they give him a little heroin, his withdrawal passes, it passes and then they don't give him heroin or methadone"* (male, 36–40, never on methadone). Additionally, the *obshchak* banned introducing young people to heroin, and such an introduction would result in immediate social consequences and often physical violence.

**Health-related concerns.** Despite the BI designed to dispel health-related myths about methadone, health-related beliefs featured prominently in qualitative interviews. Most common was the idea that methadone "eats up one's insides," leading to a protracted and painful death. It was seen as "just another drug," no better (and potentially more dangerous) than heroin. One new methadone user described the side effects when he began treatment, *"I began to lose weight. I felt weak. What else. . . my teeth started falling out. . . And plus, you're walking around like a zombie, damn it, not in your full mind"* (male, 46–50, on methadone).

Some would-be methadone program participants were also dissuaded from engaging with the program by the lack of euphoria from methadone. Substance use and associated intoxication were seen as an escape from boredom or psychological trauma. Therefore, to potentiate soporific effects of methadone and produce euphoria, some incarcerated people would combine methadone with Dimedrol (diphenhydramine) [13]. These soporific effects contributed to the misperception of methadone as harmful. *"Those who take Dimedrol, they. . . Well, it's unpleasant, you know- you're trying to have a conversation with them and they're talking nonsense, or even fall asleep"* (male, 41–45, formerly on methadone). Because of potential occupational risks (i.e., falling asleep while using a saw [33]), the *obshchak* banned Dimedrol completely. However, many people assumed that anyone who used methadone was bound to use Dimedrol eventually, especially because methadone program participants were seen as exceptionally weak-willed. As one methadone program participant explained, *"They believe that those taking methadone would not say no to Dimedrol if they are offered"* (male, 41–45, on methadone).

## Discussion

We conducted a Screening, Brief Intervention, and Referral to Treatment (SBIRT) to screen all incarcerated people within six months of release from prison in the Kyrgyz Republic and refer those with OUD to methadone treatment after a brief intervention (BI) using motivational interviewing. While nearly all participants endorsed positive attitudes toward methadone in an 8-question survey both before and after the intervention, only 3.9% of those who participated in the BI and were not already in the methadone program decided to join the program.

Some of the observed lack of uptake may be due to a ceiling effect. Twenty-three percent of study participants were already participating in the methadone program at the time of the study. It is possible that individuals who were planning on initiating methadone therapy may have already done so. Therefore, those remaining are the least likely to join the methadone program, as suggested by the low baseline methadone interest scores (Table 3). Social desirability bias, the idea that study participants may try to answer the survey in a way that they believe the researchers want to hear, may also help to explain the positive attitudes toward methadone expressed in the survey (Table 4). The research assistants conducting the

qualitative interviews were not actively promoting methadone, whereas those administering the survey questions were part of the team providing the BI to promote methadone use. Therefore, study participants may have been more likely to divulge negative feelings about methadone to the qualitative researchers rather than on the quantitative survey.

Our qualitative analysis indicated that five factors played a major role in determining the lack of methadone uptake: interpersonal relationships, interactions with the criminal justice system, logistics of taking methadone, criminal subculture, and health-related concerns. Between individuals, age played an important role in determining whether one was encouraged to take methadone. Introducing young people to drugs was an egregious crime in the within-prison subculture, as reported in previous literature [33]. Because methadone was seen as just another drug, young people were strongly discouraged from joining the methadone program, and were instead encouraged to participate in athletic activities as a way to achieve sobriety. Meanwhile, older people were seen as more set in their ways, and cultural respect for elders meant that older people were allowed or even encouraged to continue taking methadone as they had been doing. Respect for elders is part of the behavioral code that governs the prison hierarchy system [33, 34]; the phenomenon of older people with OUD being left to their own devices regarding substance use has also been reported among Israeli immigrants from the former Soviet Union [35]. Respect for elders has also been reported in other prison contexts, such as in the United States [36].

Public health has recently come to understand the concept of "risk environments:" how physical, social, economic, and policy environments precipitate and reinforce risk [37]. In this context, in terms of the social environment, people who participated in the methadone treatment program faced negative attitudes and stigma from healthcare workers, peers within the prison, and family members outside of the prison. Common themes were the social perception that those who used methadone were weak or lacked the necessary willpower to obtain sobriety, which was seen as abstaining from all substances including methadone. These ideas of sobriety and personal/community beliefs about methadone and methadone users interacted to discourage methadone uptake. Many of these beliefs regarding methadone and methadone program participants exist in North America as well, including that methadone program participation indications a lack of will power, untrustworthiness, or ongoing addiction (i.e., to methadone instead of heroin) [38, 39].

The challenging political interplay between state and citizen, also known as the "policy environment" [37], was reflected in participants' deep mistrust of the criminal justice system and the ways this mistrust figured prominently in many interactions. According to study participants, police promoted heroin use to line their own pockets and then framed people who used heroin for any crimes for which they were unable to find a culprit. Police would also often congregate near methadone clinics to arrest people on false charges. Police harassment of people who inject drugs (PWID) in the Kyrgyz Republic has been described in detail in a previous paper [40]. Similar findings have also been reported in other countries in the region, such as Azerbaijan and Ukraine [40–42], as well as in the United States, where interviewees for a newspaper article described being followed from their methadone clinics by police and arrested for minor traffic infractions [43].

In terms of the physical risk environment [37], study participants expressed concern about being tied to a daily methadone clinic, potentially limiting one's work schedule or travel abilities. Within prison, participants discussed dilution of methadone and loss of access to former services, such as heroin through the *obshchak*. Many of these logistical considerations have been previously described in other contexts; an Italian study found that methadone program participants who were allowed to take their medication at home (with certain stipulations) had significantly higher 12-month retention rates, while similar research in Vietnam found that

those with longer commutes to methadone clinics were less likely to remain in the methadone program [44, 45]. More specific to the Kyrgyz prison context, a strict within-prison hierarchy influenced many decisions regarding methadone uptake. Criminal subculture dictated who could interact with whom and whether there would be social consequences for methadone program participation. Meanwhile, heroin was used as a tool of social control within the prison [11].

Given these findings, future methadone program implementation would likely be more successful within-prison if paired with continued access to NSP, if different hierarchy statuses received methadone from different locations, and if methadone were explicitly dissociated from connections to the formal prison administration. In the community, education for family and community members about the uses and benefits of methadone might be useful to reduce stigma surrounding methadone and those who participate in the methadone program.

Limitations of this study include that upon entry into the study and during the BI, participants were made aware that this was a study designed to encourage the use of methadone. Therefore, there was strong potential for social desirability bias, given that participants likely suspected that researchers were expecting positive attitudes toward methadone. Additionally, there was significant loss to follow-up after release; only 55 of the initial 125 study participants completed six-month follow-up. Finally, previous studies of SBIRT have suggested that BI may be insufficient for engaging people in treatment or for long-term substance-use-behavior modification [46].

After performing a screening, brief intervention, and referral to treatment (SBIRT) program among people within six months of release from Kyrgyz prisons, we found that positive attitudes toward methadone did not translate into methadone uptake due to factors relating to personal relationships and stigma, distrust of the criminal justice system, logistical considerations, the specific criminal subculture within Kyrgyz prisons, and health-related concerns about methadone. Future interventions to promote methadone uptake should focus on addressing these factors, especially given the continued high incidence of hepatitis C and HIV in this vulnerable population.

## Supporting information

**S1 File. Complete list of survey questions.** Full list of survey questions asked of all participants and codebook for reading data.
(PDF)

**S2 File. Pre- and post-release interview guides.** English translation and original Russian-language interview guides used to interview study participants pre- and post-release.
(DOCX)

**S3 File. Quantitative data.** All deidentified quantitative data; S1 File may be used as a codebook for questions asked.
(XLSX)

**S1 Checklist. TREND statement checklist.**
(PDF)

**S1 Protocol.**
(PDF)

## Author Contributions

**Conceptualization:** Jaimie P. Meyer, Frederick L. Altice.

**Data curation:** Lyu Azbel.

**Formal analysis:** Amanda R. Liberman, Taylor Litz, Julia Rozanova, Lyu Azbel.

**Funding acquisition:** Amanda R. Liberman, Daniel J. Bromberg, Julia Rozanova, Jaimie P. Meyer, Frederick L. Altice.

**Investigation:** Ainura Kurmanalieva, Julia Rozanova, Lyu Azbel.

**Methodology:** Julia Rozanova, Lyu Azbel, Frederick L. Altice.

**Project administration:** Samy Galvez.

**Resources:** Ainura Kurmanalieva.

**Supervision:** Frederick L. Altice.

**Visualization:** Taylor Litz.

**Writing – original draft:** Amanda R. Liberman.

**Writing – review & editing:** Amanda R. Liberman, Daniel J. Bromberg, Ainura Kurmanalieva, Julia Rozanova, Lyu Azbel, Jaimie P. Meyer, Frederick L. Altice.

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
