## [Decision Letter · Decision Letter 0]

29 Apr 2022

PONE-D-21-24508Interest without Uptake: A Mixed-Methods Analysis of Methadone Utilization in Kyrgyz PrisonsPLOS ONE

Dear Dr. Liberman,

Thank you for submitting your manuscript to PLOS ONE. After careful consideration, we feel that it has merit but does not fully meet PLOS ONE’s publication criteria as it currently stands. Therefore, we invite you to submit a revised version of the manuscript that addresses the points raised during the review process.

The manuscript has been evaluated by two reviewers, and their comments are available below.

Please note that Reviewer 1 is mistaken regarding their first point on the clinical trials registration and appropriate checklist required. PLOS ONE's submission guidelines for clinical trials (https://journals.plos.org/plosone/s/submission-guidelines#loc-clinical-trials) have been followed correctly for this manuscript. The study design based on our criteria does require CT registration, and as a non-randomized clinical trial the TREND checklist which is included as supporting information in the manuscript PDF is the appropriate checklist to follow for this manuscript. Please do not address this point.

The reviewers have raised a number of concerns that need attention. They request additional information on methodological aspects of the study, revisions to the statistical analyses, and the presentation/discussion of results.

Could you please revise the manuscript to carefully address the concerns raised?

We look forward to receiving your revised manuscript.

Kind regards,

Sebastian Shepherd

Staff Editor

PLOS ONE

Journal Requirements:

3. In the Methods section of the manuscript, please provide additional information regarding how participants were recruited for the qualitative study, please specify whether an interview guide was used to interview the participants in your study. If yes, please describe and/or include a copy as a Supporting Information file, and finally, please consider including more information on the number of interviewers, their training and characteristics.

4. Please provide additional information regarding the considerations  made for the prisoners included in this study. For instance, please discuss whether participants were able to opt out of the study and whether individuals who did not participate receive the same treatment offered to participants.

5. Registration done retrospectively (after enrollment of participants) (TC2/PRTC Note)

Thank you for submitting your clinical trial to PLOS ONE and for providing the name of the registry and the registration number. The information in the registry entry suggests that your trial was registered after patient recruitment began. PLOS ONE strongly encourages authors to register all trials before recruiting the first participant in a study.

1) your reasons for your delay in registering this study (after enrolment of participants started);

2) confirmation that all related trials are registered by stating: “The authors confirm that all ongoing and related trials for this drug/intervention are registered.

6. During the internal evaluation of the manuscript we have noted sme discrepancies between the study protocol and the manuscript text. In particular please could you provide some clarification on the following: 

1) The protocol indicated that the study will be conducted within 7 prisons, however the manuscript text implies that 9 prisons were included. Please could you clarify whether the IRB approved this deviation. 

2) A sample size of 120 participants was calculated in the study protocol, however 125 participants were included in the study as reported in the ms text. As such please could you clarify whether the IRB approved for the inclusion of additional participants in the study.

Furthermore, please could you provide a description of the intervention of the clinical trial and please also report the expected primary and secondary outcomes of the study within the Methods section. 

Finally, please provide additional information regarding the considerations  made for the prisoners included in this study. For instance, please discuss whether participants were able to opt out of the study and whether individuals who did not participate receive the same treatment offered to participants.

7. In your Data Availability statement, you have not specified where the minimal data set underlying the results described in your manuscript can be found. PLOS defines a study's minimal data set as the underlying data used to reach the conclusions drawn in the manuscript and any additional data required to replicate the reported study findings in their entirety. All PLOS journals require that the minimal data set be made fully available. For more information about our data policy, please see http://journals.plos.org/plosone/s/data-availability.

8. Please note that in order to use the direct billing option the corresponding author must be affiliated with the chosen institute. Please either amend your manuscript to change the affiliation or corresponding author, or email us at plosone@plos.org with a request to remove this option.

9. Your abstract cannot contain citations. Please only include citations in the body text of the manuscript, and ensure that they remain in ascending numerical order on first mention.

10. We note you have included a table to which you do not refer in the text of your manuscript. Please ensure that you refer to Table 1a and 1b in your text; if accepted, production will need this reference to link the reader to the Table.

11. Please include captions for your Supporting Information files at the end of your manuscript, and update any in-text citations to match accordingly. Please see our Supporting Information guidelines for more information: http://journals.plos.org/plosone/s/supporting-information. 

Reviewers' comments:

Reviewer's Responses to Questions

**Comments to the Author**

1. Is the manuscript technically sound, and do the data support the conclusions?

Reviewer #1: Partly

Reviewer #2: Partly

2. Has the statistical analysis been performed appropriately and rigorously? 

Reviewer #1: No

Reviewer #2: I Don't Know

3. Have the authors made all data underlying the findings in their manuscript fully available?

Reviewer #1: No

Reviewer #2: Yes

4. Is the manuscript presented in an intelligible fashion and written in standard English?

Reviewer #1: Yes

Reviewer #2: Yes

5. Review Comments to the Author

Reviewer #1: My comments are as follows:

1. The manuscript has been submitted as a Clinical Trial, but a thorough reading appears that it is an observational study. However, it turns out that the study is registered in clinicaltrials.gov, with a valid NCT number. More details are needed on why the authors think that this is indeed a clinical trial. It appears there is no randomization, if I am not mistaken. If the study is claimed to be a Clinical Trial, CONSORT guidelines should be followed in reporting the results, or arguments needed on why it maybe ignored.

2. The statistical analysis plan appears mixed up with other information in the Methods section. A separate subsection is desired, which should clearly mention the tests to be used, and what would be the alternatives when standard Gaussian assumptions fail (and where paired t-tests are invalid). Relevant nonparametric method should be stated here.

3. It was really strange to see that the authors didn't provide a sample size/power statement, based on a target effect size they wanted to achieve. This would allow efficient planning to similar future studies. The sample size/power should be computed based on the primary outcome, say at 5% level, and likely one that achieves 80-90% power.

4. The study is longitudinal; it is not clear why a formal longitudinal analysis was not consucted, using mixed-effects models.

5. The analysis is a bit compromised, given that the collected data is "clustered" in nature since there are 9 prisons in total, and subjects recruited in a specific prison appear to be clustered. If one doesn't want to utilize clustered paired tests, or the alternatives, sufficient justification is necessary.

Reviewer #2: This paper addresses the important issue regarding the availability of agonist maintenance therapy to opioid addicted prisoners. Its focus on the Kyrgyz Republic is unique in view of the absence of similar information in other former Soviet Republics. It could be improved by attention to the following points:

a) describe methadone as an evidence-based treatment that reduced opioid use, risk for HIV and opioid overdose, improves overall functioning, and increases the chances for engagement in addiction and other relevant medical treatments. Describing it as the “gold standard” introduces a value judgment that is not necessary.

b) the paper focuses on prisoners who do not want methadone treatment, but it looks like about 20% of the prisoners were on it at the time of incarceration and continued it in prison. This finding needs more emphasis.

c) Can more detail be provided about the “prison hierarchy”? How do prisoners get slotted into the three categories?

d) Were the 11.2% who screened positive addicted to opioids?

e) Findings in Tables 1b and 1c can be likely be summarized in the text.

f) Any thoughts about how to increase acceptability of methadone among prisoners who are not receiving it?

6. PLOS authors have the option to publish the peer review history of their article (what does this mean?). If published, this will include your full peer review and any attached files.

Reviewer #1: No

Reviewer #2: **Yes: **George E. Woody, MD

---

## [Author Response · Author response to Decision Letter 0]

11 Aug 2022

Response: Thank you for these clarifying documents. We have updated our manuscript to reflect these formatting changes.

Response: Previously, our manuscript had included the following: “If OUD was confirmed, potential participants completed informed consent procedures in which research assistants made clear that this study was not affiliated with the prison administration, that surveys and interviews would remain anonymous, and that participants could withdraw from the study at any point.” We have updated our text to add: “Each participant was given a copy of the informed consent document to read and provided with ample time to read the form and to ask questions. To minimize potential coercion, every participant was read a statement that included the following: “Participation in the study is voluntary. Refusal or consent to participate will not affect the change of conditions of registration in prison and the information will be kept confidential. Personal information about your medical condition or any other information will be available only to you and for research purposes.” Participants were able to opt out of the study at any time and were assured that participation, refusal to participate, or discontinuation of participation were not linked with any rewards or punishments.”

3. In the Methods section of the manuscript, please provide additional information regarding how participants were recruited for the qualitative study, please specify whether an interview guide was used to interview the participants in your study. If yes, please describe and/or include a copy as a Supporting Information file, and finally, please consider including more information on the number of interviewers, their training and characteristics.

Response: Thank you for this feedback. We have modified our information about recruitment: “Briefly, the Department of Penitentiary Institutions in the Kyrgyz Republic provided the study coordinator with a list of all incarcerated persons within 180 days of scheduled release or possible early release. (Those with less than eight days until their release date were excluded.) Research personnel screened all incarcerated persons who met these criteria using a single-item screener for opioid use disorder (OUD) followed by a pre-incarceration assessment of opioid use in the 30 days before incarceration. Those who screened positive then underwent additional screening for opioid dependence using the validated Rapid Opioid Dependence Scale.”

For interview guides, English translations of pre- and post-interview guides have been included as File S2.

Additional information about interviewers was provided as follows: “LA and JR conducted the interviews in Russian, although participants were given the option to have their interviews in Kyrgyz with a trained research associate if they preferred. All participants chose to have their interviews conducted in Russian.”

4. Please provide additional information regarding the considerations made for the prisoners included in this study. For instance, please discuss whether participants were able to opt out of the study and whether individuals who did not participate receive the same treatment offered to participants.

Response: Please see response to Point 2 above. 

5. Registration done retrospectively (after enrollment of participants) (TC2/PRTC Note)

Thank you for submitting your clinical trial to PLOS ONE and for providing the name of the registry and the registration number. The information in the registry entry suggests that your trial was registered after patient recruitment began. PLOS ONE strongly encourages authors to register all trials before recruiting the first participant in a study.

1) your reasons for your delay in registering this study (after enrolment of participants started);

2) confirmation that all related trials are registered by stating: “The authors confirm that all ongoing and related trials for this drug/intervention are registered.

Response: The authors confirm that all ongoing and related trials for this drug/intervention are registered. This trial was registered in clinicaltrials.gov on July 1, 2021 which was after enrollment began. While the original intention was to register the study prior to enrollment, this delay was due to a change in staff at the beginning of the study. The timeline of enrollment and data collection for this paper’s self-reported preliminary data (2017-2021) is outlined in the results section (first sentence).

6. During the internal evaluation of the manuscript, we have noted some discrepancies between the study protocol and the manuscript text. In particular please could you provide some clarification on the following: 

1) The protocol indicated that the study will be conducted within 7 prisons, however the manuscript text implies that 9 prisons were included. Please could you clarify whether the IRB approved this deviation. 

2) A sample size of 120 participants was calculated in the study protocol, however 125 participants were included in the study as reported in the text. As such please could you clarify whether the IRB approved for the inclusion of additional participants in the study. 

Furthermore, please could you provide a description of the intervention of the clinical trial and please also report the expected primary and secondary outcomes of the study within the Methods section.

Response: We have updated our information about the intervention to include the following: “Next, [study subjects] participated in a brief intervention (BI) guided by motivational interviewing principles in which a trained research assistant explained benefits and dispelled myths relating to methadone treatment both during and after incarceration. This BI had two aims. First, the BI was designed to inform potential participants on the risks of substance misuse, abuse, and dependency by illustrating the potential hazards and adverse health consequences. Second, the BI aimed to motivate potential participants to reduce risky behavior (e.g., continued drug use) and seek treatment for their substance dependence disorder. After the BI, participants’ interest in methadone was re-assessed as above, and if they were interested, they were referred to a treating physician in the prison to initiate methadone. All participants, irrespective of methadone enrollment, underwent a second BI one week before release to encourage study participants on methadone to link to care or those not on methadone to begin methadone after release. 

During each BI, research team members provided evidence-based information on methadone. This information was available to study participants in the community upon release. Study participants were informed of the risks and benefits of methadone, and during the second BI, they learned how to access methadone in their communities. Each BI lasted approximately 20 minutes, and afterwards, participants were provided time for questions. The BIs were audio recorded; audio files are available upon request (files only available in the original languages of the BI—Russian & Kyrgyz).”

We have also added study outcomes: “Primary study outcomes included initiation of methadone, retention in methadone treatment, relapse to heroin, and recidivism.”

As for sample size, the IRB states that we will aim to recruit 120 people into the intervention, as a minimum target. Because the first step of the SBIRT procedure is S-Screening of all eligible participants, there is no way to know how many will screen positive for OUD, with 120 being a general benchmark. This sample size is allowed by the IRB.

Finally, please provide additional information regarding the considerations made for the prisoners included in this study. For instance, please discuss whether participants were able to opt out of the study and whether individuals who did not participate receive the same treatment offered to participants.

Response: Please see response to Point 2 above.

7. In your Data Availability statement, you have not specified where the minimal data set underlying the results described in your manuscript can be found. PLOS defines a study's minimal data set as the underlying data used to reach the conclusions drawn in the manuscript and any additional data required to replicate the reported study findings in their entirety. All PLOS journals require that the minimal data set be made fully available. For more information about our data policy, please see http://journals.plos.org/plosone/s/data-availability.

Response: We have uploaded the underlying quantitative data as a Supporting Information file.

8. Please note that in order to use the direct billing option the corresponding author must be affiliated with the chosen institute. Please either amend your manuscript to change the affiliation or corresponding author or email us at plosone@plos.org with a request to remove this option.

Response: The corresponding author is affiliated with Yale University, the chosen institute. These authors were unclear of the need for clarification on this point.

9. Your abstract cannot contain citations. Please only include citations in the body text of the manuscript and ensure that they remain in ascending numerical order on first mention.

Response: Thank you for this feedback. The abstract no longer contains citations.

10. We note you have included a table to which you do not refer in the text of your manuscript. Please ensure that you refer to Table 1a and 1b in your text; if accepted, production will need this reference to link the reader to the Table.

Response: Thank you for this feedback. The original manuscript referenced Table 1, without specifying Tables 1a and 1b in particular. We have updated the reference to Table 1 to specify the individual sub-tables.

11. Please include captions for your Supporting Information files at the end of your manuscript, and update any in-text citations to match accordingly. Please see our Supporting Information guidelines for more information: http://journals.plos.org/plosone/s/supporting-information. 

Response: The authors appreciate this helpful list of guidelines. Our initial manuscript did not contain Supporting Information files, but we have added these in response to reviewer/editor comments. We have added the captions at the end of the manuscript as requested.

Reviewer #1: My comments are as follows:

1. The manuscript has been submitted as a Clinical Trial, but a thorough reading appears that it is an observational study. However, it turns out that the study is registered in clinicaltrials.gov, with a valid NCT number. More details are needed on why the authors think that this is indeed a clinical trial. It appears there is no randomization, if I am not mistaken. If the study is claimed to be a Clinical Trial, CONSORT guidelines should be followed in reporting the results, or arguments needed on why it may be ignored.

Response: The authors were requested to disregard this point, per editorial staff. 

2. The statistical analysis plan appears mixed up with other information in the Methods section. A separate subsection is desired, which should clearly mention the tests to be used, and what would be the alternatives when standard Gaussian assumptions fail (and where paired t-tests are invalid). Relevant nonparametric method should be stated here.

Response: The authors appreciate this correction and have created a separate sub-section for statistical analysis. Relevant nonparametric methods have been used, as suggested.

3. It was really strange to see that the authors didn't provide a sample size/power statement, based on a target effect size they wanted to achieve. This would allow efficient planning to similar future studies. The sample size/power should be computed based on the primary outcome, say at 5% level, and likely one that achieves 80-90% power.

Response: This study tests the SBIRT method of recruiting people into OAT. During the S-Screening step, we screen all potentially eligible participants (people to be released from Kyrgyz prisons that have MMT) for opioid use disorder. Only those that screen positive are then included in the sample. In our study protocol, we present the target sample as 120 people as approximately 1000 people are released from the Kyrgyz prisons that have MMT every year, as we expected approximately 120 of them to be eligible for inclusion. Indeed, this number was fairly close as we recruited 125 people into the sample. 

For the analysis of whether the B-brief intervention component was effective, we cannot recruit more people than there are eligible participants in the country; therefore, a formal sample size calculation was not included. 

4. The study is longitudinal; it is not clear why a formal longitudinal analysis was not constructed, using mixed-effects models.

Response: In the analysis we compare OAT interest, knowledge, and attitude scores before and after the brief intervention component of SBIRT, as the goal of this analysis is to determine whether the brief intervention is successful. If we had indication that the intervention might have been successful, we would have been interested in doing a more thorough analysis to make sure that we conservative in our claims. However, the simple analysis presented in the paper did not show a significant difference in scores before/after the intervention. Therefore, we can be confident in the claims we make to the reader. 

5. The analysis is a bit compromised, given that the collected data is "clustered" in nature since there are 9 prisons in total, and subjects recruited in a specific prison appear to be clustered. If one doesn't want to utilize clustered paired tests, or the alternatives, sufficient justification is necessary.

Response: Clustered analyses e.g. clustered t-tests, mixed/multilevel models are more conservative as they account for potential additional effects from clustering. As the simple paired t-test was not found to be significant in our analysis, we can be confident that it will be even less significant if we account for additional factors like clustering. 

Reviewer #2: This paper addresses the important issue regarding the availability of agonist maintenance therapy to opioid addicted prisoners. Its focus on the Kyrgyz Republic is unique in view of the absence of similar information in other former Soviet Republics. It could be improved by attention to the following points:

a) describe methadone as an evidence-based treatment that reduced opioid use, risk for HIV and opioid overdose, improves overall functioning, and increases the chances for engagement in addiction and other relevant medical treatments. Describing it as the “gold standard” introduces a value judgment that is not necessary.

Response: The authors appreciate this feedback and have revised their original statement about methadone as follows: “Medication-assisted therapy with opioid agonists is an evidence-based HIV-prevention strategy that decreases the risk of opioid overdose, improves overall functioning, and increases engagement with other medical treatments [1–3].”

b) the paper focuses on prisoners who do not want methadone treatment, but it looks like about 20% of the prisoners were on it at the time of incarceration and continued it in prison. This finding needs more emphasis.

Response: The authors appreciate this astute observation. We have added the following: “Twenty-three percent of study participants were already participating in the methadone program at the time of the study. It is possible that individuals who were planning on initiating methadone therapy may have already done so.”

c) Can more detail be provided about the “prison hierarchy”? How do prisoners get slotted into the three categories?

Response: Thank you for this feedback. The authors have aimed to provide clarification by adding the following: “Briefly, when a person arrives into prison, he encounters a tribunal of his peers, which assesses whether he is guilty of the crime for which he has been incarcerated, the severity of his crime, and any mitigating factors (for example, positive character references from community members).”

d) Were the 11.2% who screened positive addicted to opioids?

Response: Yes; these individuals screened positive for opioid use disorder (OUD). The manuscript has been updated to clarify this point.

e) Findings in Tables 1b and 1c can likely be summarized in the text.

Response: While the authors understand the reviewer’s suggestion, we believe that these tables allow for a more in-depth portrayal of participant characteristics and provide visual comparison between those who completed and those who chose not to complete the study.

f) Any thoughts about how to increase acceptability of methadone among prisoners who are not receiving it?

Response: Thank you for this point of clarification. As we specify in the original manuscript, “Given these findings, future methadone program implementation would likely be more successful within-prison if paired with continued access to NSP, if different hierarchy statuses received methadone from different locations, and if methadone were explicitly dissociated from connections to the formal prison administration.”

---

## [Decision Letter · Decision Letter 1]

6 Sep 2022

PONE-D-21-24508R1Interest without Uptake: A Mixed-Methods Analysis of Methadone Utilization in Kyrgyz PrisonsPLOS ONE

Dear Dr. Liberman,

Thank you for submitting your manuscript to PLOS ONE. After careful consideration, we feel that it has merit but does not fully meet PLOS ONE’s publication criteria as it currently stands. Therefore, we invite you to submit a revised version of the manuscript that addresses the points raised during the review process.

Please see further comments from the reviewers below. One reviewer has has requested changes, particularly regarding clarity of the text, details regarding the prison system, and regarding how MM fits within broader negative social pressure and stigma regarding addiction. The reviewer has also suggested some opportunities to add context to this study. Please ensure you address each of the comments raised.

We look forward to receiving your revised manuscript.

Kind regards,

Hugh Cowley

Staff Editor

PLOS ONE

Reviewers' comments:

Reviewer's Responses to Questions

**Comments to the Author**

1. If the authors have adequately addressed your comments raised in a previous round of review and you feel that this manuscript is now acceptable for publication, you may indicate that here to bypass the “Comments to the Author” section, enter your conflict of interest statement in the “Confidential to Editor” section, and submit your "Accept" recommendation.

Reviewer #1: All comments have been addressed

Reviewer #2: (No Response)

2. Is the manuscript technically sound, and do the data support the conclusions?

Reviewer #1: (No Response)

Reviewer #2: Yes

3. Has the statistical analysis been performed appropriately and rigorously? 

Reviewer #1: (No Response)

Reviewer #2: I Don't Know

4. Have the authors made all data underlying the findings in their manuscript fully available?

Reviewer #1: (No Response)

Reviewer #2: (No Response)

5. Is the manuscript presented in an intelligible fashion and written in standard English?

Reviewer #1: (No Response)

Reviewer #2: No

6. Review Comments to the Author

Reviewer #1: (No Response)

Reviewer #2: Overall: This is a unique paper about addiction treatment in a part of the world where little is known about it. The use of two Russian-speaking staff to collect data likely allowed the research team to get unique information about inmate infrastructures within prisons. Outcome results are clear but the text is hard to read and could be shortened by 40-50%. Suggestions are: a) the authors mix stigma with negative attitudes about methadone maintenance (MM). Stigma applies to addiction in general; MM has its own stigma within the overall stigma of addiction. b) The negative attitudes about MM may represent what can happen when MM is started in a country where there have been laws against it for many years. I presume this is true in the Kyrgyz Republic since it likely had laws prohibiting use of opioids for treating opioid addiction when it was part of the Soviet Union. The authors might consider adding information about similar attitudes in the U.S. Examples are that MM “eats up your insides”; it “gets into your bones”; it is “just another drug”, not a rx; that people on mm are “weak”. c) The BNDD tried to arrest Dr. Dole, and it might be interesting to document it if one or more references can be found. It’s an excellent example of the very negative police response in the early day of methadone. Wyoming still may not allow MM and if so, could be an example of this lingering negativity about MM. d) Consider mentioning DSM-5 and ICD-11 where a patient can be in remission if on MM, buprenorphine, or naltrexone. e) Consider mentioning that MM is available in prisons in most EU countries. Modify the statement that the Kyrgyz Republic is one of “few countries”. e) Add a comment on how SBIRT has shown weak to modest effectiveness in the U.S. and not studies on former Soviet States. f) can more details be added about how new arrivals are classified within the prison system. The system the authors describe is unique. g) translations of Russian names relevant here. h) Is the prison infrastructure re methadone use common to all prisons? If so, could its description be shortened? i) what is the relationship between prison management and the way prisoners are classified? It appears as if prisoners are put into the groupings by other prisoners. j) does the MOH put out info on MM? k) consider limiting the number of English translations of Russian words; repeating the same words makes the paper more difficult to read l) It’s nor surprising that few started MM in view of the negative social pressure about it. Comments along those lined could be added since this negativity may explain why so few who were not on methadone started it.

7. PLOS authors have the option to publish the peer review history of their article (what does this mean?). If published, this will include your full peer review and any attached files.

Reviewer #1: No

Reviewer #2: No

---

## [Author Response · Author response to Decision Letter 1]

8 Sep 2022

1. Outcome results are clear but the text is hard to read and could be shortened by 40-50%. 

The authors thank the reviewer for this suggestion. This version of the text has been revised and shortened from 8742 words to 7098 words. 

2. The authors mix stigma with negative attitudes about methadone maintenance (MM). Stigma applies to addiction in general; MM has its own stigma within the overall stigma of addiction. 

The authors have applied the reviewer’s suggestion and have edited the section about stigma to clarify.

3. The negative attitudes about MM may represent what can happen when MM is started in a country where there have been laws against it for many years. I presume this is true in the Kyrgyz Republic since it likely had laws prohibiting use of opioids for treating opioid addiction when it was part of the Soviet Union. The authors might consider adding information about similar attitudes in the U.S. Examples are that MM “eats up your insides”; it “gets into your bones”; it is “just another drug”, not a rx; that people on mm are “weak”. 

The authors thank the author for this suggestion and have added the following sentence (lines 366-369): “Many of these beliefs regarding methadone and methadone program participants exist in North America as well, including that methadone program participation indications a lack of will power, untrustworthiness, or ongoing addiction (i.e., to methadone instead of heroin) [42,43].”

4. The BNDD tried to arrest Dr. Dole, and it might be interesting to document it if one or more references can be found. It’s an excellent example of the very negative police response in the early day of methadone. Wyoming still may not allow MM and if so, could be an example of this lingering negativity about MM. 

While the authors were not able to find a reference regarding the attempted arrest of Dr. Dole, they have added a source regarding police harassment of MM patients in the United States (lines 378-380): “Similar findings have also been reported… in the United States, where interviewees for a newspaper article described being followed from their methadone clinics by police and arrested for minor traffic infractions [47].”

5. Consider mentioning DSM-5 and ICD-11 where a patient can be in remission if on MM, buprenorphine, or naltrexone. 

The authors thank the reviewer for this suggestion. However, they have chosen not to include these given the differing definitions of remission in the DSM-5-TR vs the ICD-11. The ICD-11 defines “sustained full remission” as “After a diagnosis of Opioid dependence, and often following a treatment episode or other intervention (including self-intervention), the person has been abstinent from opioids for 12 months or longer (Source: ICD-11 for Mortality and Morbidity Statistics (who.int)). Meanwhile, the DSM-5-TR allows methadone patients to receive a diagnosis of early or sustained remission, as long as the specifier “on maintenance therapy” is added (Source: Psychiatry Online | DSM Library). Given these differing definitions, we have chosen not to mention either definition of remission in this text.

6. Consider mentioning that MM is available in prisons in most EU countries. Modify the statement that the Kyrgyz Republic is one of “few countries”. 

We thank the reviewer for this suggestion, and we have modified the statement in question (line 29). 

7. Add a comment on how SBIRT has shown weak to modest effectiveness in the U.S. and not studies on former Soviet States.

We thank the reviewer for this suggestion, and we have modified lines 67-70 to reflect this point.

8. can more details be added about how new arrivals are classified within the prison system. The system the authors describe is unique. 

As described in lines 276-280, “Briefly, when a person arrives into prison, he encounters a tribunal of his peers, which assesses whether he is guilty of the crime for which he has been incarcerated, the severity of his crime, and any mitigating factors (for example, positive character references from community members). He is then assigned a hierarchy status based on this assessment (Table 1b).”

9. translations of Russian names relevant here. 

The authors have attempted to remove as many of the Russian transliterations as possible to encourage ease of reading.

10. Is the prison infrastructure re methadone use common to all prisons? If so, could its description be shortened? 

As described in the text (line 274), this infrastructure is common to men’s prisons in the Kyrgyz Republic. Its description has been shortened as requested.

11. what is the relationship between prison management and the way prisoners are classified? It appears as if prisoners are put into the groupings by other prisoners.

To answer the reviewer’s question, incarcerated people are put into the groupings by their peers, as described in lines 276-280. This is an informal prison governmental system, not affiliated with the official prison administration (more details about this relationship have been described in a previous paper, Liberman et al. 2021).

12. does the MOH put out info on MM? 

The authors thank the reviewer for this question, and have added a note about MOH guidelines for MM (lines 57-59). 

13. consider limiting the number of English translations of Russian words; repeating the same words makes the paper more difficult to read 

The authors thank the reviewer for this suggestion, and have attempted to eliminate as many English transliterations of Russian words as possible to promote ease of reading throughout the text.

14. It’s not surprising that few started MM in view of the negative social pressure about it. Comments along those lined could be added since this negativity may explain why so few who were not on methadone started it.

The authors thank the reviewer for this suggestion, highlighted in lines 365-367: “These ideas of sobriety and personal/community beliefs about methadone and methadone users interacted to discourage methadone uptake.”

---

## [Decision Letter · Decision Letter 2]

26 Sep 2022

PONE-D-21-24508R2Interest without Uptake: A Mixed-Methods Analysis of Methadone Utilization in Kyrgyz PrisonsPLOS ONE

Dear Dr. Liberman,

Thank you for submitting your manuscript to PLOS ONE and for your patience during this review process as there have been a number of transitions in the handling of this manuscript. We invite you to submit a revised version of the manuscript that addresses the points raised during the review process.  As you can see there are just a few additional requests/clarifications that have been raised by the reviewers. It is my belief that these final responses/clarifications will result in a manuscript that is acceptable for publication and that the research will be an important contribution to the community of Plos One readers. 

We look forward to receiving your revised manuscript.

Kind regards,

Judith I Tsui

Academic Editor

PLOS ONE

Journal Requirements:

Reviewers' comments:

Reviewer's Responses to Questions

**Comments to the Author**

1. If the authors have adequately addressed your comments raised in a previous round of review and you feel that this manuscript is now acceptable for publication, you may indicate that here to bypass the “Comments to the Author” section, enter your conflict of interest statement in the “Confidential to Editor” section, and submit your "Accept" recommendation.

Reviewer #1: (No Response)

Reviewer #2: (No Response)

2. Is the manuscript technically sound, and do the data support the conclusions?

Reviewer #1: Partly

Reviewer #2: Yes

3. Has the statistical analysis been performed appropriately and rigorously? 

Reviewer #1: No

Reviewer #2: N/A

4. Have the authors made all data underlying the findings in their manuscript fully available?

Reviewer #1: No

Reviewer #2: Yes

5. Is the manuscript presented in an intelligible fashion and written in standard English?

Reviewer #1: No

Reviewer #2: Yes

6. Review Comments to the Author

Reviewer #1: The manuscript has been submitted as a Clinical Trial! I was surprized to see that there is no mention of a formal sample size/power calculation, and the study data has been analyzed, without any perspective on that. That needs to be added, or justification provided on why that is missing!

Reviewer #2: This is a much-improved manuscript. A few suggestions for additional improvements are: 1) Can the authors provide details about heroin availability in the prisons? In the U.S. heroin distribution would not be tolerate. How is this situation managed in the prisons? It appears to be an informal distribution network that is tolerate by prison administrators. Do all the guards participate or only a few? More details would be interesting if possible. 2) How is methadone treatment administered in the prisons? Do prisoners come to medical units for daily observed dosing? Is methadone brought to them by medical staff every day? 3) Where is the NSP located and how to prisoners access it?

7. PLOS authors have the option to publish the peer review history of their article (what does this mean?). If published, this will include your full peer review and any attached files.

Reviewer #1: No

Reviewer #2: **Yes: **George E. Woody, MD. Emeritus Professor, Department of Psychiatry. Perelman School of Medicine at the University of Pennsylvania

---

## [Author Response · Author response to Decision Letter 2]

6 Oct 2022

1. The manuscript has been submitted as a Clinical Trial! I was surprized to see that there is no mention of a formal sample size/power calculation, and the study data has been analyzed, without any perspective on that. That needs to be added, or justification provided on why that is missing!

The authors thank the reviewer for this suggestion. This was addressed in a previous round of reviews (7.27.22), and the original comment and response are provided below. Additionally, the text has been updated to incorporate the answer provided below:

“Comment: It was really strange to see that the authors didn't provide a sample size/power statement, based on a target effect size they wanted to achieve. This would allow efficient planning to similar future studies. The sample size/power should be computed based on the primary outcome, say at 5% level, and likely one that achieves 80-90% power.

Response: This study tests the SBIRT method of recruiting people into OAT. During the S-Screening step, we screen all potentially eligible participants (people to be released from Kyrgyz prisons that have MMT) for opioid use disorder. Only those that screen positive are then included in the sample. In our study protocol, we present the target sample as 120 people as approximately 1000 people are released from the Kyrgyz prisons that have MMT every year, as we expected approximately 120 of them to be eligible for inclusion. Indeed, this number was fairly close as we recruited 125 people into the sample.

For the analysis of whether the B-brief intervention component was effective, we cannot recruit more people than there are eligible participants in the country; therefore, a formal sample size calculation was not included.”

2. Can the authors provide details about heroin availability in the prisons? In the U.S. heroin distribution would not be tolerate. How is this situation managed in the prisons? It appears to be an informal distribution network that is tolerate by prison administrators. Do all the guards participate or only a few? More details would be interesting if possible.

The authors thank the reviewer for this suggestion. They have added the following text: “The obshchak was highly motivated to dissuade people from using methadone, because the obshchak was the major distributer of in-prison heroin. This process was facilitated by the obshchak's extensive connections outside of the prison. It acted both as a mutual aid fund, collecting and redistributing goods to incarcerated people, and also as a marketplace for various goods and services. This marketplace was facilitated by corruption of official prison staff, who allowed these goods, including heroin, to enter the prison [11]. Heroin served both as a commodity and as a form of currency which could be used to purchase other items within the prison.” 

It is difficult to estimate a percentage of official prison staff who participate in the informal heroin distribution system, so this specific detail was not included.

3. How is methadone treatment administered in the prisons? Do prisoners come to medical units for daily observed dosing? Is methadone brought to them by medical staff every day? 

The authors thank the reviewer for this suggestion and have added the following: “Prison medical staff administered methadone daily at a specific, designated location for all methadone program participants irrespective of hierarchy level. Therefore, joining the methadone program meant potential physical interaction with people of lower hierarchy status or using the same items, like pens or cups. These interactions could lead to demotion within the hierarchy [29].”

4. Where is the NSP located and how to prisoners access it?

The authors thank the reviewers for this clarification point. The specific location of the NSP depends on the prison; some are located in the medical area, whereas other are in the barracks. Participants register confidentially in the program, and there is a low threshold for entry. Nurses distribute the clean equipment. Although the NSP is not a central focus of this paper, we have clarified this in the text with the following: “Ordinarily, to access the NSP, individuals registered confidentially and were then able to receive injection equipment from nurses located either in the medical area or the barracks, depending on the specific prison. However, those that joined the methadone program were no longer permitted to participate in the NSP.”

---

## [Editor Report · Decision Letter 3]

13 Oct 2022

Interest without Uptake: A Mixed-Methods Analysis of Methadone Utilization in Kyrgyz Prisons

PONE-D-21-24508R3

Dear Dr. Liberman,

We’re pleased to inform you that your manuscript has been judged scientifically suitable for publication and will be formally accepted for publication once it meets all outstanding technical requirements.

Kind regards,

Judith I Tsui

Academic Editor

PLOS ONE
---

## [Editor Report · Acceptance letter]

17 Oct 2022

PONE-D-21-24508R3 

Interest without Uptake: A Mixed-Methods Analysis of Methadone Utilization in Kyrgyz Prisons 

Dear Dr. Liberman:

I'm pleased to inform you that your manuscript has been deemed suitable for publication in PLOS ONE. Congratulations! Your manuscript is now with our production department. 

Kind regards, 

on behalf of

Dr. Judith I Tsui 

Academic Editor

PLOS ONE